# Ethanol Extract of the Microalga *Phaeodactylum tricornutum* Shows Hepatoprotective Effects against Acetaminophen-Induced Acute Liver Injury in Mice

**DOI:** 10.3390/ijms25116247

**Published:** 2024-06-06

**Authors:** Dae Yoon Kim, Hui Jin Park, Jae-In Eom, Cheol-Ho Han, Cheol-Ho Pan, Jae Kwon Lee

**Affiliations:** 1College of Pharmacy, Chungbuk National University, Cheongju 28160, Republic of Korea; lnbsky@naver.com; 2Department of Biology Education, College of Education, Chungbuk National University, Cheongju 28160, Republic of Korea; glwldl@naver.com; 3Microalgae Ask Us Co., Ltd., Gangneung 25441, Republic of Korea; umjaein@maus2020.com (J.-I.E.); hancheolho@maus2020.com (C.-H.H.)

**Keywords:** acute liver failure, hepatotoxicity, *Phaeodactylum tricornutum*, acetaminophen, D-galactosamine

## Abstract

Acute liver failure is an infrequent yet fatal condition marked by rapid liver function decline, leading to abnormalities in blood clotting and cognitive impairment among individuals without prior liver ailments. The primary reasons for liver failure are infection with hepatitis virus or overdose of certain medicines, such as acetaminophen. *Phaeodactylum tricornutum* (PT), a type of microalgae known as a diatom species, has been reported to contain an active ingredient with anti-inflammatory and anti-obesity effects. In this study, we evaluated the preventive and therapeutic activities of PT extract in acute liver failure. To achieve our purpose, we used two different acute liver failure models: acetaminophen- and D-GalN/LPS-induced acute liver failure. PT extract showed protective activity against acetaminophen-induced acute liver failure through attenuation of the inflammatory response. However, we failed to demonstrate the protective effects of PT against acute liver injury in the D-GalN/LPS model. Although the PT extract did not show protective activity against two different acute liver failure animal models, this study clearly demonstrates the importance of considering the differences among animal models when selecting an acute liver failure model for evaluation.

## 1. Introduction

Microalgae offer a promising alternative source of valuable nutrients, including fatty acids and proteins [1]. Cultured microalgae have yielded a diverse range of over 15,000 isolated compounds, encompassing fatty acids, sterols, phenolic compounds, terpenes, enzymes, polysaccharides, alkaloids, toxins, and pigments such as lutein and β-carotene [2]. *Phaeodactylum tricornutum* (PT), a type of microalgae known as a diatom species, has been reported to contain a significant amount of eicosapentaenoic acid, reaching up to 35% of its total fatty acid content [3]. On a dry-weight basis, it consists of approximately 36.4% protein, 26.1% carbohydrates, 18.0% lipids, 15.9% ash, and 0.25% neutral detergent fiber [4]. PT has gained attention as a promising producer of fucoxanthin, as its fucoxanthin content can be enriched up to one hundred-fold more than that for brown seaweeds [5,6]. Due to its unique structure as a carotenoid compound, fucoxanthin serves two primary biological functions: quenching singlet oxygen and scavenging free radicals [7]. Additionally, scientific research has indicated that fucoxanthin exhibits promising effects in mitigating hepatic injury in a mouse model of non-alcoholic steatohepatitis (NASH), induced by a choline-deficient diet. Its administration led to a reduction in hepatic fat accumulation, liver weight gain, hepatic lipid oxidation, and inflammation associated with NASH by effectively inhibiting the production of chemokines [8].

Acute hepatic failure (AHF) is associated with an alarmingly high mortality rate of 80% [9]. Many studies have been conducted on animal models of acute hepatic failure, but implementing any model in practice is not an easy task [10,11]. Each model typically focuses on a specific aspect of AHF, lacking a comprehensive representation of the syndrome as a whole. Therefore, it is challenging to clearly establish the hepatoprotective and therapeutic effects of test substances in a single type of AHF animal model. In this study, we experimented with two types of AHF models—the acetaminophen model and the D-galactosamine/lipopolysaccharide (D-GalN/LPS) model—to investigate the preventive and therapeutic effects of PT extract on AHF.

Over the last forty years, a significant amount of research has been conducted on the mechanism of acetaminophen (APAP) hepatotoxicity using murine models. APAP undergoes bioactivation in the liver, primarily by the enzyme cytochrome 2E1 (CYP2E1) and, to a lesser extent, cytochrome 1A2 (CYP1A2). This bioactivation process results in the formation of a highly reactive and toxic metabolite called N-acetyl-p-benzoquinone imine (NAPQI). NAPQI leads to the depletion of glutathione (GSH) levels within the liver and subsequently binds to proteins, triggering oxidative stress, mitochondrial dysfunction, and, ultimately, necrotic cell death [12,13]. The acute co-injection of D-GalN/LPS has become a commonly employed experimental model for studying acute liver failure. D-GalN/LPS-induced hepatotoxicity is related to an increased inflammatory response and the generation of reactive oxygen species [14,15]. The liver is particularly susceptible to the effects of LPS, and the addition of D-GalN significantly exacerbates the detrimental outcome caused by LPS [16].

The current study was designed to evaluate the preventive and therapeutic activity of PT in acute liver failure. To achieve our purpose, we used two different acute liver failure models: acetaminophen- and D-GalN/LPS-induced. This study clearly demonstrates the importance of considering the differences among pharmacological models when selecting an acute liver failure model to evaluate hepatoprotective and therapeutic active compounds.

## 2. Results

### 2.1. Liver Injury Protection Effect of PT Extract against Acetaminophen-Induced Acute Liver Failure

To explore the preventive (Figure 1A,B) and therapeutic (Figure 1C,D) effects of PT extract in acute liver failure, we first used an acute liver failure mouse model induced by acetaminophen. The severity of acetaminophen-induced liver injury was evaluated by measuring the serum liver enzyme level at 24 h after acetaminophen injection. The prevention experiments were performed via pre-treatment with PT extract for 10 days before acetaminophen injection. As shown in Figure 1A,B, serum glutamic oxaloacetic transaminase (GOT) and serum glutamic pyruvic transaminase (GPT) levels significantly increased after the acetaminophen injection. However, these scores decreased by almost half by the PT extract, in a manner that was independent of injection concentration (100 or 200 mg/kg). In this experiment, *Hovenia dulcis* Thunb. extract (HD) and *Silybum marianum* seed extract (SM) were used as positive controls, where the protective effect of SM against acute liver failure was superior to that of HD. For acute liver failure, PT showed a better effect when used for preventive application than therapeutic application.

### 2.2. Liver Injury Protection Effect of PT Extract against D-GalN/LPS-Induced Acute Liver Failure

Next, the preventive (Figure 2A,B) and therapeutic (Figure 2C,D) effects of PT extract on acute liver failure were confirmed in another acute liver failure model induced by D-GalN/LPS. The prevention experiments (Figure 2A,B) were performed voa pre-treatment with PT extract for 3 days before D-GalN/LPS injection. D-GalN/LPS-induced liver toxicity was evaluated by measuring GOT and GPT levels at 6 h after D-GalN/LPS injection. In the therapeutic experiments (Figure 2C,D), D-GalN/LPS was injected 6 h before the oral administration of the PT extract. After oral administration of the PT extract, an autopsy was performed 6 h later. In D-GalN/LPS model animal experiments, the effect of PT on acute liver injury did not alleviate hepatotoxicity in both preventive and therapeutic experiments. As shown in Figure 2, GOT and GPT levels increased following D-GalN/LPS injection, but these levels were reduced through injection of the positive control drug, *Hovenia dulcis* Thunb. extract (HD). However, the GOT and GPT levels which increased following D-GalN/LPS injection were not decreased with the administration of PT extract.

### 2.3. Preventive Effect of PT on Acetaminophen- or D-GalN/LPS-Induced Liver Injury Models

The preventive effect of PT on acute liver injury was then investigated through histopathological examination. The histopathological examination of the liver was conducted after H&E staining. As shown in Figure 3A, the livers of the control group (normal) mice showed normal histology of the liver. Acetaminophen injection at the tested doses induced apparent morphological changes, including a loss of normal hepatic architecture, hepatic sinusoid hyperemia, congestion of the sinusoids, and focal damage around the central vein. Pre-treatment with control extracts (*Hovenia dulcis* Thunb. extract, HD, and *Silybum marianum* Seed extract, SM) decreased hepatic sinusoid hyperemia and congestion of the sinusoids and recovered hepatic architecture to a normal shape. The PT extract also showed similar preventive effects as the control extracts. Briefly, pre-treatment with PT extract attenuated the hyperemia and congestion of the sinusoids. These results indicated that the PT extract had preventive effects on APAP-induced acute liver failure.

To represent the numerical measurement of sinusoidal and central vein congestion, we used an accurate and simple method in ImageJ analysis^®^ software (V1.8.0). As shown in Figure 3B,C, the extravasation of red blood cells (RBCs) between the sinusoid and central vein increased with the injection of acetaminophen, and the extravasation of RBCs associated with APAP decreased following the oral administration of the positive control drugs and PT extract. These results indicate that endothelial damage to the central vein and sinusoidal hemorrhage were reduced through treatment with PT extract. Figure 3C represents the extravasation of RBCs between the sinusoid and the central vein in 200× magnification view, as a number calculated using ImageJ. These results indicate that even 100 μg/mL of PT had a similar effect to the positive control.

D-GalN/LPS injection also induced a loss of normal hepatic architecture, hepatic sinusoid hyperemia, congestion of the sinusoids, and focal damage around the central vein (Figure 4A). Pre-treatment with *Hovenia dulcis* Thunb. extract (HD) decreased hepatic sinusoid hyperemia and congestion of the sinusoids and recovered hepatic architecture to a normal shape. However, pre-treatment with PT extract showed no or weak preventive effects on hepatic morphological changes. Even when high concentrations of PT were administered, a weak protective effect was observed against D-GalN/LPS-induced liver tissue damage.

### 2.4. Expression Levels of Inflammatory Cytokines in Acute Liver Failure Mice

To investigate the anti-inflammatory effects of PT in acute liver failure mice, the levels of the cytokines TNF-α and IL-1β in sera, which are representative of inflammation, were determined using enzyme-linked immunosorbent assays (ELISA). Cytokine levels were measured in sera obtained from the preventive experiments for acetaminophen (Figure 5A,B) and D-GalN/LPS (Figure 5C,D) models. As shown in Figure 5A,B, TNF-α and IL-1β levels significantly increased following acetaminophen injection; however, these levels decreased in a statistically significant manner through pre-treatment with the control drugs (SM extract) and PT extracts (100 or 200 mg/kg). D-GalN/LPS also enhanced TNF-α and IL-1β production (Figure 5C,D); however, unlike the acetaminophen model, pre-treatment with PT did not inhibit the production of TNF-α and IL-1β in the D-GalN/LPS model.

## 3. Discussion

The aim of this study was to demonstrate the impact of PT extract on acute liver failure and associated metabolic pathways in acute liver injury model animals. PT is a potentially valuable dietary supplement, as PT-derived carotenoids are anti-inflammatory and antioxidant compounds [17]. The most representative pigment derived from PT is fucoxanthin, which has been proven to possess various pharmacological effects [6,17]. In addition, PT extract also showed therapeutic effects in animal models of obesity-related metabolic disorders [18]. *Hovenia dulcis*, traditionally utilized for its hangover relief and liver health benefits, is under investigation for its potential therapeutic applications in treating non-alcoholic steatohepatitis (NASH), a severe manifestation of non-alcoholic fatty liver disease (NAFLD) characterized by excessive fat accumulation, inflammation, and liver damage. Therefore, we used *Hovenia dulcis* Thunb. extract as a positive control [19]. In this study, we considered two different acute liver injury models—induced with acetaminophen and D-GalN/LPS, respectively—and most of the positive results were obtained with the acetaminophen model.

Acute liver failure carries an extremely high mortality rate of 80% [9]. Viral hepatitis is the most frequent cause of acute liver failure in the world, followed increasingly closely by chemicals (e.g., drugs and toxins). At the therapeutic dose, acetaminophen is one of the safest and most effective medicines [20]; however, an overdose of acetaminophen can cause liver injury and failure, resulting in a significant number of emergency visits and hospitalizations [21,22]. The acetaminophen model is well established for analyzing the recovery effect of acute liver failure in animal models through the administration of candidate compounds [23]. The concentration of 600 mg/mL of acetaminophen used in this study was sufficient to induce acute liver failure and led to consistent results within the experimental groups.

Very complicated mechanisms are involved in the pathogenesis of acetaminophen-induced acute liver disease. These mechanisms encompass the conversion of cytochrome P450 metabolism into a reactive metabolite that depletes glutathione and forms covalent bonds with proteins. Concurrently, a reduction in glutathione levels accompanied by augmented generation of reactive oxygen and nitrogen species in hepatocytes occurs [24]. This leads to heightened oxidative stress, which is associated with disruptions in calcium homeostasis and the initiation of signal transduction responses, ultimately resulting in mitochondrial permeability transition. Moreover, this mitochondrial permeability transition takes place concomitantly with increased oxidative stress, decline in mitochondrial membrane potential, and impairment of ATP synthesis within the mitochondria. Ultimately, the depletion of ATP culminates in necrosis [12,13]. Although mitochondrial oxidative stress is a major reason for hepatocyte necrosis, there is a discrepancy between early GSH depletion and delayed necrosis [25]. As a result, it has been hypothesized that the initial oxidative stress alone is not adequate to induce hepatocyte necrosis, and an additional trigger is required to intensify this oxidant stress. The additional trigger that is necessary to enhance this oxidant stress appears to be mediated by the activation (phosphorylation) of the mitogen-activated protein kinase (MAPK) [26]. For these reasons, we plan to conduct an experiment on the signaling pathways of MAPK.

TNF-α has been associated with elevated oxidative stress, including increased generation of reactive oxygen species and reactive nitrogen species, as well as the recruitment and activation of other inflammatory cells [27]. Blazka et al. reported significant increases in the serum levels of TNF-α and IL-1α in mice treated with acetaminophen [28]. Moreover, anti-TNF-α or anti-IL-1α antibodies partially prevented hepatotoxicity in acetaminophen-intoxicated mice [29]. In our experimental findings, we observed an elevation in the production of inflammatory cytokines—specifically TNF-a and IL-1—in a mouse model of acetaminophen-induced acute liver failure. However, the preventive treatment with PT (especially the high dose) determined a decrease in the levels of these cytokines.

In this study, we also tested another acute liver failure model—using D-GalN/LPS—in order to confirm the preventive and therapeutic effects of PT in acute liver failure. The D-GalN/LPS model has also been used as an animal model for simulating the formation of acute liver failure in humans [30]. D-GalN can consume uridine monophosphate in hepatocytes, resulting in the depletion of nucleic acid and damage to hepatocyte structure and function, finally causing the apoptosis of cells [31]. LPS is a cell wall component of Gram-negative bacteria. High concentrations of LPS overcome the detoxification process and cause obvious liver damage [32]. LPS combination models with D-GalN, alcohol, or concanavalin A for acute liver failure are useful for demonstrating blood chemical profiles and metabolic changes. These changes start generally within 2–4 h but return to normal values after 12 h if the model animal survives. Moreover, a weak point of the D-GalN/LPS model is that animals may die between 4–8 h before the autopsy if even a small amount of D-GalN/LPS over the safe dose is injected. In our experiment, the D-GalN model showed excessively high hepatotoxic levels (ALT and AST) or the animals died before the analysis of serum profiles; therefore, this model made it difficult to verify the hepatoprotective effect of PT compared to the acetaminophen model.

## 4. Materials and Methods

### 4.1. Materials and Reagents

Acetaminophen, D-Galactosamine, and Lipopolysaccharide were purchased from Sigma-Aldrich (St. Louis, MO, USA). *Hovenia dulcis* Thunb. extract (HD) was purchased from Nutra Green Biotechnology Co., Ltd. (Shanghai, China) and *Silybum marianum* seed extract (SM) was purchased from Naturex (Avignon, France). The fucoxanthin standard was purchased from Sigma-Aldrich (St. Louis, MO, USA). Unless indicated, all other chemicals were also purchased from Sigma-Aldrich.

### 4.2. Culture of Phaeodactylum tricornutum (PT)

The PT (UTEX-646) used in this study was obtained from the University of Texas at Austin (UTEX) algal culture collection. The PT was cultured using F/2+Si medium and seawater. The composition of the F/2+Si medium was as follows: NaNO_3_ 75 mg, NaH_2_PO_4_ 5.65 mg, Na_2_SiO_3_ 30 mg, Na_2_EDTA 4.16 mg, FeCl_3_ 3.15 mg, CuSO_4_ 0.01 mg, ZnSO_4_ 0.022 mg, CoCl_2_ 0.01 mg, MnCl_2_ 0.18 mg, Na_2_MoO_4_ 0.006 mg, vitamin B12 0.0005 mg, vitamin B1 0.1 mg, and biotin 0.0005 mg. All media were sterilized at 121 °C for 15 min before use. Seawater was sterilized with sodium hypochlorite (NaOCl) in a 50-ton photobioreactor (PBR), and F/2+Si stock solution was added, inoculated with PT, and incubated for approximately 7 days. At the end of incubation, PT biomass was harvested using a continuous centrifuge.

### 4.3. Preparation of PT Extract

Wet PT biomass was extracted through stirring for 15 h with the addition of 95% ethanol to achieve a final alcohol concentration of 70%. It was decompression filtered through filter paper (Whatman, pore size 5 μm, Bucks, UK), mixed with excipients (β-cyclodextrin; ES Food Ingredients, Gunpo, Republic of Korea) using a homogenizer (13,000 rpm for 5 min), and concentrated (40 °C, 100 mbar, 100 rpm) using a vacuum rotary evaporator (Rotavapor R-300, BUCHI, Flawil, Switzerland). The concentrated extract was sterilized at 70 °C for 30 min and freeze dried.

### 4.4. Characterization of the PT Extract

To standardize the PT extract, lead, arsenic, cadmium, mercury, and *Escherichia coli*, as well as the content of fucoxanthin, were analyzed in the PT extract. All analyses were conducted by the Korea Health Supplement Institute (Seoul, Republic of Korea)—a certified testing laboratory designated by the Ministry of Food and Drug Safety (MFDS, Osong, Republic of Korea)—according to the general test methods for standards and specifications of Korean food. Analysis results for three lots of PT extract were as follows: fucoxanthin content at 18.14, 17.31, and 18.33 mg/g (Figure 6); lead (Pb) content at 0.0313, 0.0312, and 0.0323 ppm; arsenic (As) content at 0.0124, 0.0152, and 0.0168 ppm; cadmium (Cd) content at 0.0344, 0.0368, and 0.0375 ppm; and mercury (Hg) content at 0.0047, 0.0050, and 0.0047 ppm. The established standards and specifications for the PT extract are defined as follows: the concentration of fucoxanthin should range from 14.33 to 21.50 mg/g. The lead (Pb) concentrations must be below 1.0 ppm, arsenic (As) must be below 1.0 ppm, cadmium (Cd) must be below 0.3 ppm, and mercury (Hg) must be below 0.5 ppm. *Escherichia coli* must not be detectable in any of the samples. Additionally, the total pheophorbide content should not exceed 1000 ppm.

### 4.5. Analysis of Fucoxanthin

Fucoxanthin was analyzed using HPLC with an Agilent 1260 system (Agilent Technologies, Santa Clara, CA, USA), according to previous methods [33]. The analysis utilized a CAPCELL PAK C18 MG II column (particle size 5 μm, 250 × 4.6 mm I.D.; Phenomenex, CA, USA). The mobile phase consisted of acetonitrile (A) and water (B), with a flow rate of 1 mL/min. The gradient elution program started with a composition of 90:10 (A:B), increased to 100:0 over 8 min, maintained at 100:0 for 3 min, and then decreased to 80:20 over the following 5 min. Detection was performed at 450 nm. Quantification of fucoxanthin was achieved by measuring peak areas and comparing them to a calibration curve (1, 5, 10, 50, 100, and 200 μg/mL) using a fucoxanthin standard.

### 4.6. Animals

Seven-week-old male C57BL/6J mice weighing 21.0 ± 3.0 g were purchased from Dooyeol Biotech (Seoul, Republic of Korea). The mice were fed with a standard chow diet and water freely. They were housed under normal laboratory conditions (23 ± 2 °C, 12 h light–dark cycle) with continuous access to food and water. All experiments were conducted under the standard procedure set by the Committee for the Purpose of Control and Supervision of Experiments on Animals and the National Institutes of Health for the specification use of the experimental animals. The experimental protocol was approved by the Ethics Committee for Animal Experimentation of Chungbuk National University (Permit Number: CBNUA-633-13-01, Republic of Korea).

### 4.7. Acetaminophen-Induced Acute Liver Injury Model

The C57BL/6J mice were divided into 6 groups (each *n* = 8), as shown in Table 1.

Acute liver injury using acetaminophen was induced as described in detail previously [34,35]. Briefly, for the study of PT prevention activity (Figure 7A), C57BL/6J mice were administered PT, SM, or HD orally for 10 days. Then, the animals received 600 mg/kg of APAP (Sigma Aldrich) through intraperitoneal injection after fasting overnight for a period of approximately 15–16 h. After 24 h, the mice were sacrificed, their blood was collected for serum separation, and their livers were removed and stored at −80 °C. Sera were separated from the blood for aspartate transaminase (AST) and alanine transaminase (ALT), and the remainder of the sera was stored at −80 °C for further use. In case of PT therapeutic activity (Figure 7B), the animals received 600 mg/kg of APAP (Sigma Aldrich) through intraperitoneal injection after fasting overnight, and PT, SM, or HD was administrated orally for 3 days. After 24 h, the mice were sacrificed, their blood was collected, and their livers were removed. Sera were separated from the blood for AST (GOT) and ALT (GPT) level determination, and the remainder of the sera was stored at −80 °C for further use. A portion of each liver was fixed in 10% neutral buffer formalin for hematoxylin and eosin (H&E) staining which allows for the visualization of the structure and distribution of cells and morphological changes within a tissue sample, and the remainder of the liver tissue was snap-frozen and stored at −80 °C for protein analyses.

### 4.8. D-GalN- and LPS-Induced Acute Liver Injury Model

D-GalN (450 mg/kg) and LPS (10 μg/kg) were dissolved in sterile 0.9% sodium chloride, according to the product description. The mice were divided into 6 groups (each *n* = 8) randomly, as shown in Table 2.

To test the acute liver toxicity prevention ability of PT (Figure 7C), C57BL/6J mice were administered PT, SM, or HD orally for 3 days. Then, they received D-GalN/LPS (Sigma Aldrich) via intraperitoneal injection after fasting overnight for approximately 15–16 h. After 6 h, the mice were sacrificed, their blood was collected for serum separation, and their livers were removed and stored at −80 °C. In the case of PT therapeutic activity (Figure 7D), the animals received D-GalN/LPS (Sigma Aldrich) via intraperitoneal injection after fasting overnight for approximately 15 h. After 24 h, the mice were sacrificed, their blood was collected, and their livers were removed. Sera were separated from the blood for GOT and GPT level determination, and the remainder of the sera was stored at −80 °C for further use.

### 4.9. Histological Analysis

Formalin-fixed livers were embedded in paraffin. The sections (4 μm thick) were stained with hematoxylin and eosin (H&E) solution (American Histolabs, Gaithersburg, MD, USA) for histological analysis. Stained liver tissues were examined under a light microscope (Olympus, Tokyo, Japan). The numerical measurement of sinusoidal and central vein congestion was analyzed using ImageJ analysis^®^ software. To quantify the extravasation of red blood cells (RBCs) in an image of a mouse liver tissue section stained with H&E, the tissue image was analyzed through the following four steps: (1) change the scale to micrometers; (2) convert the image to grayscale; (3) segment (isolate) the dark-red-stained RBCs using thresholding; and (4) measure the thresholded area.

### 4.10. Enzyme Immunoassay

The concentrations of TNF-α and IL-1β (in sera were measured through enzyme-linked immunosorbent assays (ELISA; R&D Systems, Minneapolis, MN, USA), according to the manufacturer’s instructions.

## 5. Conclusions

This study provides reasonable evidence that PT effectively protects hepatocytes against acetaminophen-induced acute liver failure by attenuating the inflammatory response. These findings provide valuable insights into the potential of PT as a medicinal intervention for preventing acute liver failure. Although this study failed to demonstrate the protective effects of PT against acute liver injury in the D-GalN/LPS model, future efforts are planned to establish an optimal model by varying the concentrations of D-GalN and LPS.

## Figures and Tables

**Figure 1 ijms-25-06247-f001:**
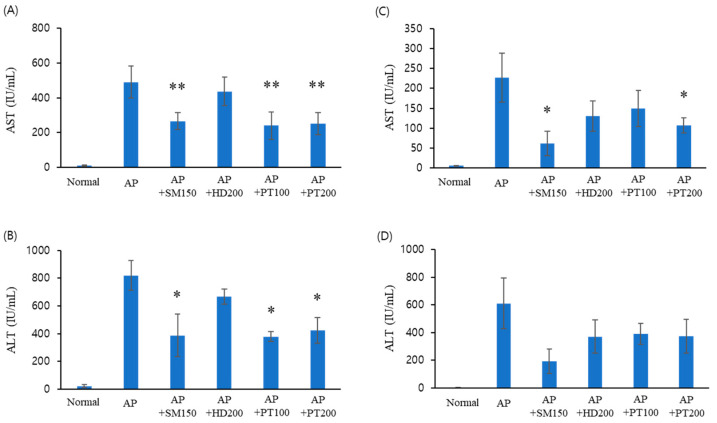
The preventive (**A**,**B**) and therapeutic (**C**,**D**) effects of PT extract on acute liver injury after acetaminophen overdose. (**A**,**B**) C57BL/6J mice were administered SM, HD, and PT orally for 10 days, and then, the animals received 600 mg/kg of intraperitoneal acetaminophen after fasting overnight. After 24 h, the mice were sacrificed, and blood was collected for serum separation. Sera were separated from blood for AST (**A**) and ALT (**B**), where these acronyms refer to aspartate transaminase and alanine transaminase, respectively. (**C**,**D**) Mice received acetaminophen via intraperitoneal injection after fasting overnight and then administered PT orally for 3 days. The data from three independent experiments, each of which was performed in triplicate, are indicated as the mean ± SD. The significance of the differences between the value of the AP group and that of the SM, HD, and PT treatment groups after AP injection were determined through one-way ANOVA with Tukey’s comparisons test: * *p* < 0.05 and ** *p* < 0.01.

**Figure 2 ijms-25-06247-f002:**
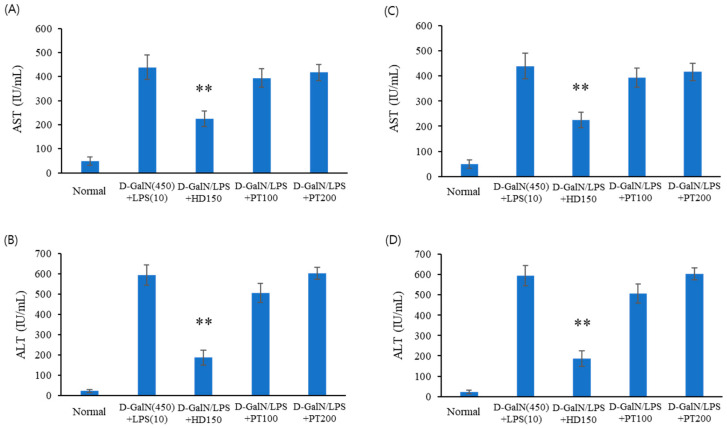
The preventive (**A**,**B**) and therapeutic (**C**,**D**) effects of PT extract on acute liver injury after D-GalN/LPS injection. (**A**,**B**) C57BL/6J mice were administered HD and PT orally for 3 days and then received intraperitoneal D-GalN (450 mg/kg) and LPS (10 μg/kg) after fasting overnight. After 6 h, the mice were sacrificed, and their blood was collected for serum separation. Sera were separated from blood for AST (**A**) and ALT (**B**). (**C**,**D**) The mice received D-GalN and LPS via intraperitoneal injection after fasting overnight and were then administered HD and PT orally. After 6 h, the mice were sacrificed, and their blood was collected for serum separation. Sera were separated from blood for AST (**C**) and ALT (**D**). The data from three independent experiments, each of which was performed in triplicate, are indicated as the mean ± SD. The significances of the differences between the value of the D-GalN/LPS group and that of the HD and PT treatment groups after D-GalN/LPS injection were determined through one-way ANOVA with Tukey’s comparisons test: ** *p* < 0.01.

**Figure 3 ijms-25-06247-f003:**
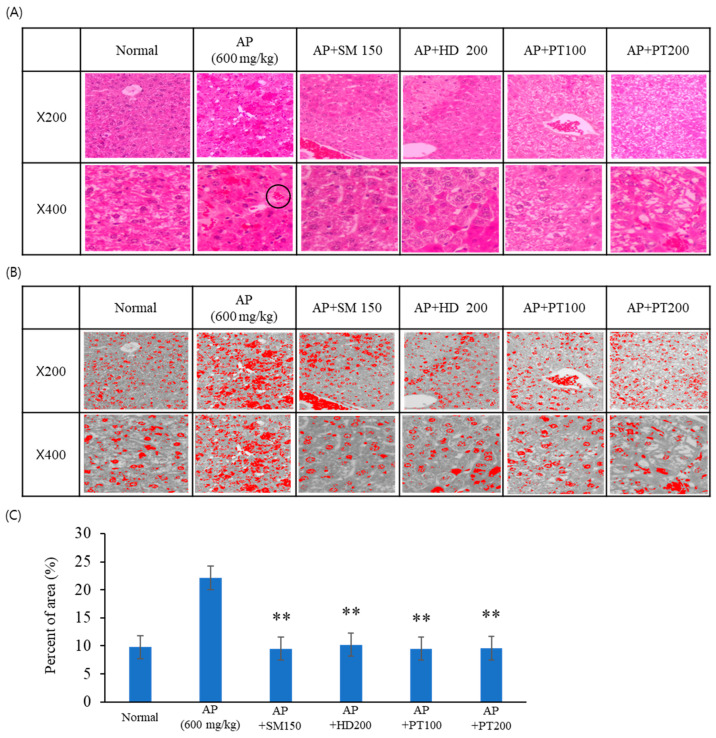
Histological analysis of liver tissues. Liver tissue sections of normal and acetaminophen-injected mice stained with H&E are shown (magnification, 200× and 400×). (**A**) H&E staining images; (**B**) grayscale images converted with ImageJ; (**C**) percentage of area of extravasated RBCs in the image. The black circle in (**A**) indicates accumulated RBCs. The significance of the differences between the value of the AP group and that of the SM, HD, and PT treatment groups after AP injection was determined through one-way ANOVA with Tukey’s comparisons test: ** *p* < 0.01.

**Figure 4 ijms-25-06247-f004:**
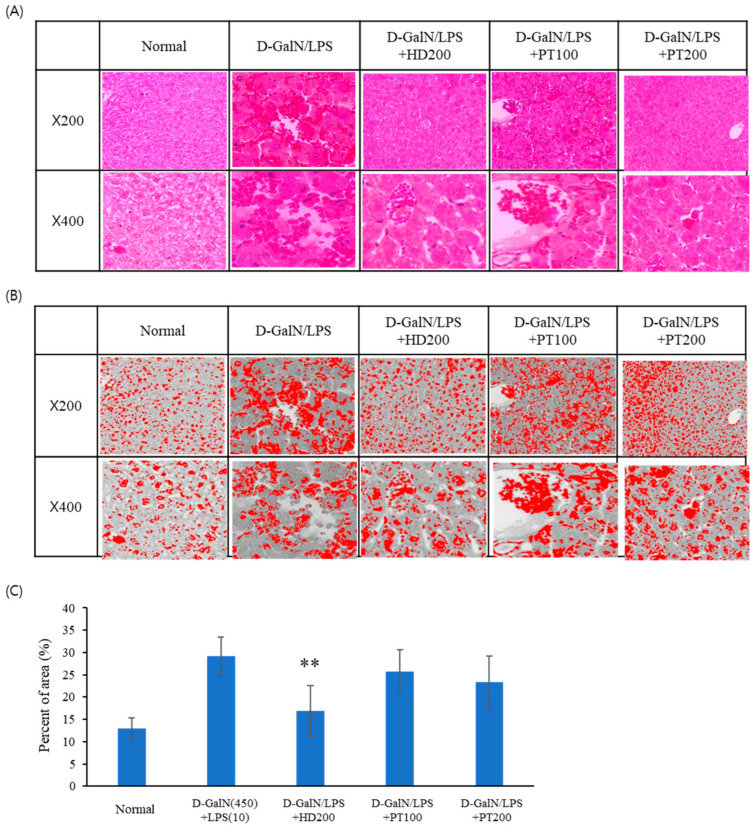
Histological analysis of liver tissues. Liver tissue sections of normal and D-GalN/LPS-injected mice stained with H&E are shown (magnification, 200× and 400×). (**A**) H&E staining images; (**B**) grayscale images converted with ImageJ; (**C**) percentage of area of extravasated RBCs in the image. The significance of the differences between the value of the D-GalN/LPS group and that of the HD and PT treatment groups after D-GalN/LPS injection were determined through one-way ANOVA with Tukey’s comparisons test: ** *p* < 0.01.

**Figure 5 ijms-25-06247-f005:**
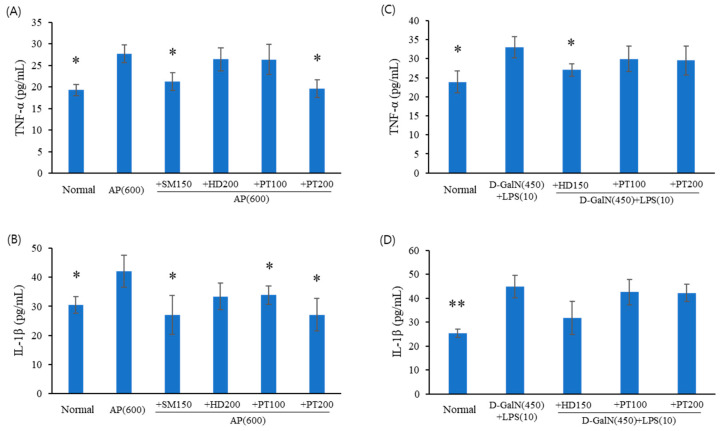
Effect of PT on the production of inflammatory cytokines in acute liver failure mouse models. Sera obtained from acetaminophen (**A**,**B**) and D-GalN/LPS (**C**,**D**) animal models were used in ELISA assays for TNF-α and IL-1β. The data from three independent experiments, each of which was performed in triplicate, are indicated as the mean ± SD. The significance of the differences between the value of AP or D-GalN/LPS groups and that of the SM, HD, and PT treatment groups after AP or D-GalN/LPS injection were determined through one-way ANOVA with Tukey’s comparisons test: * *p* < 0.05 and ** *p* < 0.01.

**Figure 6 ijms-25-06247-f006:**
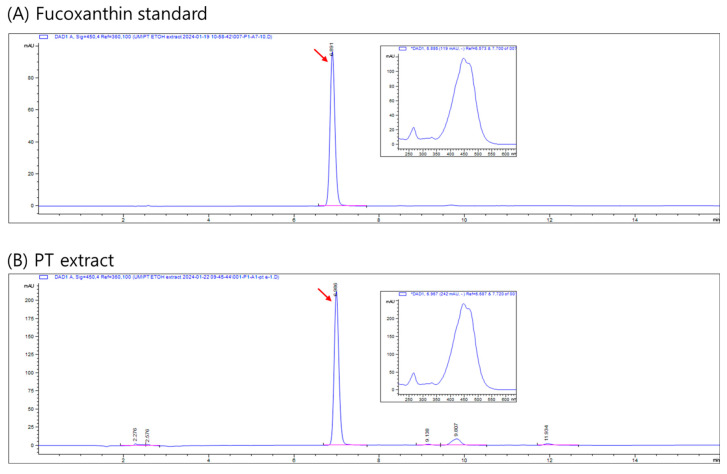
HPLC chromatogram of fucoxanthin standard (**A**) and PT extract (**B**) at 450 nm. The inset shows the absorption spectrum of the fucoxanthin peak. Red arrows indicate the fucoxanthin peak in each chromatogram.

**Figure 7 ijms-25-06247-f007:**
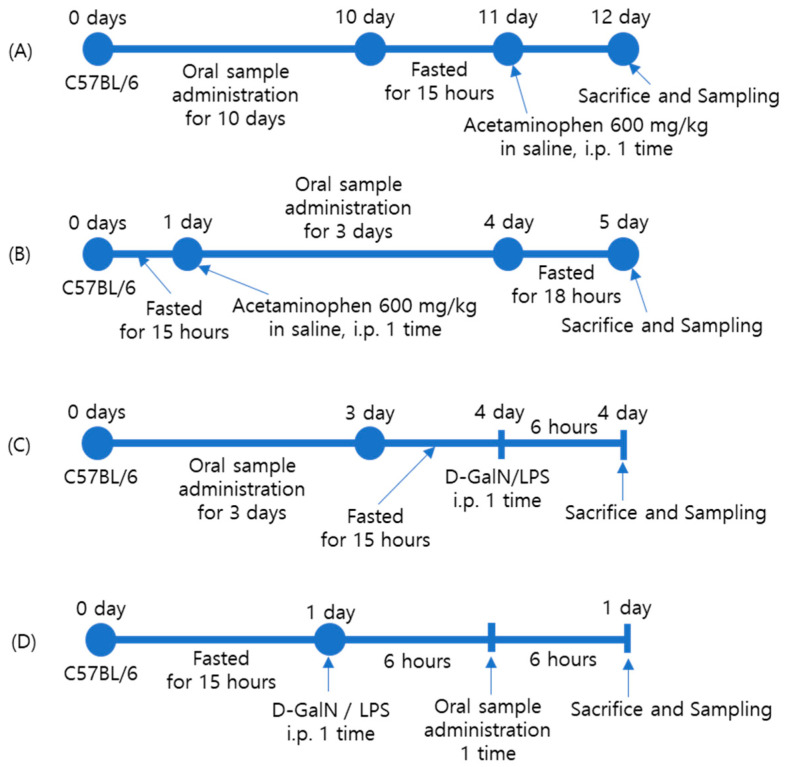
Outline of the experimental design for the in vivo study of the preventive (**A**) and therapeutic (**B**) effects of PT extract on acetaminophen-induced acute liver injury and the preventive (**C**) and therapeutic (**D**) effects of PT extract on D-GalN/LPS-induced acute liver injury.

**Table 1 ijms-25-06247-t001:** Experimental groups in acetaminophen-induced acute liver injury model.

Number	Group	Explanation
1	Normal	Untreated control group, without AP injection
2	AP	600 mg/kg AP injection (IP)
3	AP + SM	Oral administration of 150 mg/kg *Silybum marianum* Seed extract after AP injection.
4	AP + HD	Oral administration of 200 mg/kg *Hovenia dulcis* Thunb. extract after AP injection.
5	AP + PT100	Oral administration of 100 mg/kg PT extract after AP injection.
6	AP + PT200	Oral administration of 200 mg/kg PT extract after AP injection.

**Table 2 ijms-25-06247-t002:** Experimental groups in D-GalN/LPS—induced acute liver injury model.

Number	Group	Explanation
1	Normal	Untreated control group, without D-GalN/LPS injection
2	D-GalN/LPS	IP injection of D-GalN/LPS
3	D-GalN/LPS + SM	Oral administration of 150 mg/kg *Silybum marianum* Seed extract after D-GalN/LPS injection.
4	D-GalN/LPS + HD	Oral administration of 200 mg/kg *Hovenia dulcis* Thunb. extract after D-GalN/LPS injection.
5	D-GalN/LPS + PT100	Oral administration of 100 mg/kg PT extract after D-GalN/LPS injection.
6	D-GalN/LPS + PT200	Oral administration of 200 mg/kg PT extract after D-GalN/LPS injection.

## Data Availability

The corresponding authors can make any materials available upon request. The aggregate data from the referenced data sets are available from the corresponding authors upon reasonable request.

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
