# Peer review of "Ethanol Extract of the Microalga Phaeodactylum tricornutum Shows Hepatoprotective Effects against Acetaminophen-Induced Acute Liver Injury in Mice"

_ijms, 2024, doi:10.3390/ijms25116247_

Round 1

Reviewer 1 Report

Comments and Suggestions for Authors

1.Phaeocystis tricornutum (PT), a type of microalgae known as diatoms, contains an active compound with anti-inflammatory and anti-obesity effects. This study evaluated the preventive and therapeutic activities of PT extracts against acute liver failure. 2.Two different models of acute liver failure were used: acetaminophen or D-GalN/LPS. PT extract has a protective effect against paracetamol-induced acute liver failure by reducing the inflammatory response. The protective effect of PT on acute liver injury could not be demonstrated in the D-GalN/LPS model. 3.This study provides experimental basis for the use of Phaeophyta tricornutum to treat acute liver injury, and has certain application prospects. 4.It is recommended that the full text be published after revision.

1.This experiment used Hovenia dulcis extract (HD) and Milk Thistle Seed extract (SM) were used as positive controls, but the source could not be provided. 2.Page 10, line 290: What are the sources fucoxanthin standard? 3.Is the Quantitative methods verified of fucoxanthin? 4.The discussion part of the article only discusses and analyzes the research results of Phaeocystis tricornutum, and there is no discussion for the Hovenia dulcis extract (HD) which is effective in both models. 5.In the reference section, the last two references 37-38 are not cited in the text.

Author Response

The authors would like to thank the reviewer for your useful comments that improved the quality of our manuscript. Based on the comments of the reviewer, the manuscript has been amended as follows. Modified parts are marked in red for easy identification.

(Reviewer 1)

It is recommended that the full text be published after revision.

  1. This experiment used Hovenia dulcis extract (HD) and Milk Thistle Seed extract (SM) were used as positive controls, but the source could not be provided.

(Response) We have added supply sources for HA and SM (Lines 266-268).

  1. Page 10, line 290: What are the sources fucoxanthin standard?

(Response) We have added it (Lines 268-269).

  1. Is the Quantitative methods verified of fucoxanthin?

(Response) We wrote subchapter 4.5 on fucoxanthin analysis. And reference 33 was cited, which includes a previously established fucoxanthin assay (Lines 304-313).

  1. The discussion part of the article only discusses and analyzes the research results of Phaeocystis tricornutum, and there is no discussion for the Hovenia dulcis extract (HD) which is effective in both models.

(Response) A discussion on this matter has been added along with reference 19 (Lines 202-206).

  1. In the reference section, the last two references 37-38 are not cited in the text.

(Response) All references were carefully reviewed and references 9 to 35 have been modified.

Thank you for your consideration of our manuscript.

Sincerely,

Cheol-Ho Pan

Reviewer 2 Report

Comments and Suggestions for Authors

The title: Ethanol extract of the microalga Phaeodactylum tricornutum inhibits hepatotoxicity against acute liver injury induced by acetaminophen or D-galactosamine/lipopolysaccharide (D-GalN/LPS) in mice

The title does not reflect the obtained results. According to the presented results, the ethanol extract of the microalga Phaeodactylum tricornutum showed protective effects only when using the acetaminophen model, and not the D-GalN/LPS model. Based on these, the authors go further and underline the importance of proper selection of the pharmacological models when assessing the hepatoprotective effects in acute liver failure. Therefore, a more accurate title could be (only as a suggestion):

“The importance of pharmacological model selection for acute liver failure: the example of Phaeodactylum tricornutum ethanolic extract”

Or, closer to your results:

“Ethanol extract of the microalga Phaeodactylum tricornutum shows hepatoprotective effects in acetaminophen-induced acute liver injury in mice”

Line 16 – “or take a drug” – please reformulate and revise English

Line 17- “an active compounds” – please revise English

Line 18 – “the prevent...activity” – please revise the English, “ of the PT extract on”

Line 23 – “was not showed” – please revise English

Line 24 – “that the importance of...” - please revise English

Introduction

Line 36 – reference [3] must be placed inside a cited phrase/sentence.

Line 38 – “since its' content” – please revise English

Line 49 – References [10] and [11] have nothing to do with various animal models used to study AHF. These articles are related to emergency liver transplantation in humans. Also, as the references are from 1994 and 1995 – they cannot describe the models used « over the past three decades » – line 48

Lines 52-57 – the text is repeating, please correct.

Line 53 – Please explain the abbreviation D-GalN/LPS

Line 65 – there is no need to use 5 references (12 to 16) to explain the well-known formation of acetaminophen’s toxic metabolites.

Line 71 – “the prevent and therapeutic activity” – please revise English

Line 74 – “the differences among animal models” – there are differences among the pharmacological models, not among the animal models. Both groups used mice, the differences were made by the pharmacological interventions to induce acute live injuries.

Of the 38 total references, 19 are used in the introduction section. In my opinion, many of them are added extra, there is no need to use two or more references for stating generally well-known facts.

Also, in the introduction part, you should state the previous studies which explored the hepatoprotective effects of the microalga Phaeodactylum tricornutum, their main results, their limitations, and from here, to propose a different approach, another point of view brought by this study.

Examples of previous studies worth mentioning:

·         Fidel, J., & Rodríguez, E. (2022). Phaeodactylum tricornutum as Fucoxanthin Biofactory Model and Hepatoprotective Effect of Encapsulated Spirulina and Fucoxanthin. Applied Sciences, 13(13), 7794. https://doi.org/10.3390/app13137794

·         Claire Mayer, Martine Côme, Lionel Ulmann, Graziella Chini Zittelli, Cecilia Faraloni, et al.. Preventive Effects of the Marine Microalga Phaeodactylum tricornutum, Used as a Food Supplement, on Risk Factors Associated with Metabolic Syndrome in Wistar Rats. Nutrients, 2019, 11 (5), pp.1069. ff10.3390/nu11051069ff. ffhal-02495591f

·         Kang, MJ., Kim, S.M., Jeong, SM. et al. Antioxidant effect of Phaeodactylum tricornutum in mice fed high-fat diet. Food Sci Biotechnol 22, 107–113 (2013). https://doi.org/10.1007/s10068-013-0015-y

(or some of these studies can be used in the discussion part, to correlate your results with previous results).

Materials and Methods

Line 257 – Lipopolysaccharide is repeating

Lines 263 - 265 – Please write NaNO3 and the rest of the medium salts with subscripts (like this NaNO3 )

Line 278 – “are as follows” replace with “were as follows”

Lines 278- 286 – I would separate the characterization of the extract by a new subchapter  4.4. « Characterization of the extract » (or similar), where I would offer more details about the HPLC methods used, how lead, arsenic, cadmium, and mercury content were determined (now it is not specified), how the established standards and specifications for PT extract have been defined.

Line 304-307 – perhaps using a table to present the six groups would be clearer.

Line 313 – please explain AST and ALT abbreviations

Lines 315-316 – correct – “the animals received….” and “PT, SM and HD were administrated orally for 3 days.”

Here, if PT, SM, and HD were administrated separately, please replace with “and PT, SM or HD was administrated orally for 3 days »

Line 320 – correct – a portion of each liver was fixed….

Lines 327 -332 - perhaps using a table to present the six groups would be clearer.

Line 345 – please explain « H&E solution »

Line 349 – « was analyzed the following four » please revise English

Please place figure 7 after the subchapter 4.6

Results

Line 78  - “To explore the prevention...” replace with “To explore the preventive...”

Line 81 – enzymes instead of enzyme

Line 91 – “The protective (A-B)....” should be replaced with “The preventive (A-B)...”

Lines 84-85, 96, 103 – please revise English

Lines 109-115 – please improve English fluency and the ease of reading

Line 117 – “The protective (A-B)....” should be replaced with “The preventive (A-B)...”

Line 120 - please revise the English

Line 131 – please replace “normal mice” with “control group”

Lines 135, 138 – please replace “control drugs” with “control extracts”

Lines 140-141 – “These results showed that PT extracts have therapeutic effects on APAP-induced acute liver failure” – the paragraph above describes the preventive effects of PT in acute liver failure (PT administrated before the hepatotoxicity-inducing agents) and not about its therapeutic effects.

Line 176, 177 - please revise the English

Lines 179 – 181 – “As shown in Fig. 5 (A) and (B), TNF-α and IL-1β levels were significantly increased by acetaminophen injection, but these levels were decreased by almost half levels by pretreatment of control drugs (HD or SM extract) and PT extracts (100 or 200 mg/kg). – if you examine Fig 5A and Fig 5B – the decrease of TNF – alpha and IL-1 following various extracts treatments is not close to 50% of AP group.

Lines 189-190 – “The significance of the difference between the value of Normal and that of SM, HD, and PT treated was determined by one-way ANOVA with Tukey’s comparisons test as * P < 0.05, ** P < 0.01. » - Here, the statistics should identify of PT treatment has brought significant changes compared to AP group (if there is a, indeed, a protective effect of the PT treatment).

Discussion

Line 197 – models

Line 195 – Reference 20 concerns the composition of another microalga – Nannochloropsis

Lines 202-204 – please revise the English

Line 205 – The acetaminophen model…the recovery effect….

Line 209 – « ….active molecule ». The study did not identify the active molecule responsible for the hepatoprotective effects.

Line 210 – “A very complicated mechanisms” – please correct

Line 221 – “is major reason for necrosis of hepatocyte,” – is a major reason for hepatocyte necrosis

Lines 226-227 – please correct English

Lines 235 -236 – the preventive treatment with PT (especially in the high dose) determined the decrease of those cytokines.

Lines 237, 239 – correct grammar

Line 224 – what is con A? please explain

Line 245 – please correct English (is make useful demonstrating…)

Lines 245-247 – the sentence is repeating

Lines 249-254 – please revise English.

How the D-GalN/LPS doses were selected? Is is possible that the PT extract would manifest hepatoprotective effects if the D-GalN/LPS doses were lower? Why the fact that some mice died before the analysis of serum profiles was not mentioned in the results?

Line 416 - Reference [9] is not correctly cited in the References list. Please correct.

Line 498 – Reference [33] is about the melatonin effects in the human liver, not about the hepatotoxicity of D-GalN

Lines 505-510 – References [35], [36] should be replaced by a single, more relevant, actual, reference. 

Comments on the Quality of English Language

The English Language generally requires extensive editing. There are paragraphs very well-written, but there are also paragraphs that need a close revision. 

Author Response

Dear Reviewer,

Best regards,

Cheol-Ho Pan

Round 2

Reviewer 2 Report

Comments and Suggestions for Authors

Dear authors, Thank you for taking into account my suggestions. After corrections, I have identified only some minor English errors: 

Line 259 - "die may" - please correct

Line 260 - use the past tense - "was injected"

Apart from these changes, the manuscript can be published as is. 

Comments on the Quality of English Language

Minor corrections of the English language. 

Author Response

Dear Reviewer,

The authors would like to thank the reviewers for their helpful comments that improved the quality of the manuscript. The authors have revised the manuscript as follows according to Reviewer 2's opinion. Modified areas are marked in blue for easy identification.

  1. Line 259 - "die may" - please correct

(Response) We have modified to “may die” (Line 258).

  1. Line 260 - use the past tense - "was injected"

(Response) We have have modified to “was injected” (Line 259).

  1. Apart from these changes, the manuscript can be published as is. 

(Response) Thank you for your thorough and detailed review.

Thank you for your consideration of our manuscript.

Sincerely,

Cheol-Ho Pan
